# Gene Expression of Transient Receptor Potential Channels in Peripheral Blood Mononuclear Cells of Inflammatory Bowel Disease Patients

**DOI:** 10.3390/jcm9082643

**Published:** 2020-08-14

**Authors:** Taku Morita, Keiichi Mitsuyama, Hiroshi Yamasaki, Atsushi Mori, Tetsuhiro Yoshimura, Toshihiro Araki, Masaru Morita, Kozo Tsuruta, Sayo Yamasaki, Kotaro Kuwaki, Shinichiro Yoshioka, Hidetoshi Takedatsu, Takuji Torimura

**Affiliations:** 1Department of Medicine, Division of Gastroenterology, School of Medicine, Kurume University, 67 Asahi-Machi, Kurume 830-0011, Japan; morita_taku@med.kurume-u.ac.jp (T.M.); yamasaki_hiroshi@kurume-u.ac.jp (H.Y.); mori_atsushi@med.kurume-u.ac.jp (A.M.); yoshimura_tetsuhiro@med.kurume-u.ac.jp (T.Y.); araki_toshihiro@med.kurume-u.ac.jp (T.A.); morita_masaru@med.kurume-u.ac.jp (M.M.); tsuruta_kouzou@med.kurume-u.ac.jp (K.T.); yasumoto_sayo@med.kurume-u.ac.jp (S.Y.); kuwaki_koutarou@kurume-u.ac.jp (K.K.); yoshioka_shinichirou@kurume-u.ac.jp (S.Y.); takedatsu_hidetoshi@med.kurume-u.ac.jp (H.T.); tori@med.kurume-u.ac.jp (T.T.); 2Inflammatory Bowel Disease Center, Kurume University Hospital, 67 Asahi-Machi, Kurume 830-0011, Japan

**Keywords:** crohn’s disease, mononuclear cells, transient receptor potential channel, ulcerative colitis

## Abstract

We examined the expression profile of transient receptor potential (TRP) channels in peripheral blood mononuclear cells (PBMCs) from patients with inflammatory bowel disease (IBD). PBMCs were obtained from 41 ulcerative colitis (UC) patients, 34 Crohn’s disease (CD) patients, and 30 normal subjects. mRNA levels of TRP channels were measured using the quantitative real-time polymerase chain reaction, and correlation tests with disease ranking, as well as laboratory parameters, were performed. Compared with controls, TRPV2 and TRPC1 mRNA expression was lower, while that of TRPM2, was higher in PBMCs of UC and CD patients. Moreover, TRPV3 mRNA expression was lower, while that of TRPV4 was higher in CD patients. TRPC6 mRNA expression was higher in patients with CD than in patients with UC. There was also a tendency for the expression of TRPV2 mRNA to be negatively correlated with disease activity in patients with UC and CD, while that of TRPM4 mRNA was negatively correlated with disease activity only in patients with UC. PBMCs from patients with IBD exhibited varying mRNA expression levels of TRP channel members, which may play an important role in the progression of IBD.

## 1. Introduction

Inflammatory bowel disease (IBD), consisting primarily of ulcerative colitis (UC) and Crohn’s disease (CD), represents a group of chronic inflammatory disorders involving the gastrointestinal tract. Its pathogenesis is complex; however, current research suggests that an intricate network of multiple interacting mechanisms in genetically susceptible individuals may orchestrate the dysregulation of immune and inflammatory responses [1,2].

The transient receptor potential (TRP) channel family, a diverse family of proteins that are reportedly expressed throughout the human body, have emerged as a novel and interrelated system to detect and respond to various environmental stimuli including mechanical, thermal, or chemical stimuli. With this function, they are likely to be sensors for monitoring specific responses to different exogenous and endogenous chemical and physical stimuli [3,4,5,6]. Members of the TRP superfamily are classified according to their amino acid sequences and structural similarities into canonical (TRPC), vanilloid (TRPV), melastatin (TRPM), ankyrin (TRPA), polycystin (TRPP), and mucolipin (TRPML) subfamilies [3,4,5]. It was recently revealed that the role of TRP channels reaches beyond the control of neuropeptide release from sensory nerves as they are also expressed in immune and resident tissue cells, such as epithelial cells, where they modulate many functions including cytokine production and migration. Therefore, a vital interplay between these cells appears to maintain homeostasis in the intestine, while the disruption of this interplay may be involved in the development and maintenance of IBD [7,8,9,10].

In the gut wall of IBD patients, immune cells migrating from systemic circulation release various inflammatory and immunoregulatory molecules, such as cytokines and free radicals, that modulate intestinal inflammation and tissue damage [11]. Therefore, peripheral blood mononuclear cells (PBMCs) are of particular interest due to their involvement in inflammatory and immune cell function. PBMCs may be in contact with multiple stimuli within the blood that have the potential to release TRP channels, resulting in modulation of intestinal inflammation. Therefore, an evaluation of TRP channel expression in PBMCs from IBD patients may provide insights regarding the disease pathogenesis; while changes in their expression level may mirror intestinal inflammation. Recently, the expression of these TRP channel members in PBMCs have been reported in healthy subjects and patients with various diseases [12,13,14,15]. However, to date no data regarding their expression in PBMCs from patients with IBD has been reported.

TRPV1–TRPV4; TRPM2, TRPM4, and TRPM5; and TRPC1 as well as TRPC3–TRPC7, are the most relevant TRP channels in immune cells under normal and specific disease conditions [12,13,14,15]. Therefore, this study focused on the expression profiles of these TRP channel members in PBMCs obtained from patients with IBD. Furthermore, the relationships between TRP channel levels and the disease activity index, as well as other laboratory parameters were investigated.

## 2. Materials and Methods

### 2.1. Ethical Considerations

This study was approved by the Ethical Committee of Kurume University (14253, 25 March 2015). Written informed consent was obtained from each subject before enrollment in the study.

### 2.2. Patients

This study was conducted at Kurume University Hospital between September 2011 and May 2016. A total of 105 subjects, including 41 patients with UC, 34 with CD, and 30 healthy subjects were enrolled. Prior to commencing the study, 19 subjects were excluded due to an insufficient sample volume. The patient diagnosis was based on characteristic clinical, endoscopic, radiological, and histological features. The patient characteristics and the medical therapy they received are summarized in Table 1.

### 2.3. Clinical Evaluations

For the evaluation of disease activity, clinical activity in patients with UC was graded using the partial Mayo score (PMS), with the inactive disease defined as a score ≤2, with no individual sub-score >1 point [16]. Patients with CD were graded according to the CD activity index (CDAI) comprised of eight factors, each added after adjustment with a weighted factor, with the inactive disease defined as a score <150 points [17].

### 2.4. Determination of Laboratory Parameters

Blood samples were collected from all patients and were used to measure the following laboratory parameters: Total leukocyte count, serum levels of hemoglobin, albumin, and C-reactive protein (CRP).

### 2.5. Separation of PBMCs and RNA Extraction

Blood samples (10 mL) were obtained by cubital venous puncture and collected in standard sterile polystyrene vacuum tubes with heparin. First, freshly drawn blood was diluted at a ratio of 1:2.5 with a phosphate buffered saline. PBMCs were isolated from the diluted blood by a Ficoll-Paque (GE Healthcare, Uppsala, Sweden) density gradient centrifugation according to the manufacturer’s instructions. PBMCs were pelleted, snap-frozen on dry ice, and stored at −80 °C until use [18]. RNA was extracted from PBMC samples following the protocol described for the TRIzol reagent (Invitrogen, Carlsbad, CA, USA). The quantity and purity of the RNA were determined for all samples on a Nanodrop ND-1000 spectrophotometer (Thermo Scientific, Waltham, MA, USA). The average yield was 23,000 ng. The purity, as measured by the A260/280 ratio, was between 1.91 and 1.95 [18].

### 2.6. Measurement of TRP Channel mRNA Expression Using Real-Time Quantitative Polymerase Chain Reaction (Real-Time qPCR)

Total RNA was converted into cDNA using the ReverTra Ace qPCR RT kit (Toyobo, Osaka, Japan). The generated cDNAs (25 ng) were stored at −20 °C. cDNA was added to the TaqMan Gene Expression Master Mix (Applied Biosystems, Foster City, CA, USA). qPCR reactions (20 μL) composed of 2 μL cDNA template, TaqMan Universal PCR Master Mix (2×, Thermo Fisher Scientific, Foster City, CA, USA), TaqMan assay (20×, Thermo Fisher Scientific), and H₂O. RT-PCR was performed using the StepOne Real-Time PCR System (Applied Biosystems). Reactions, run in triplicate, were incubated at 50 °C for 2 min and 95 °C for 10 min, followed by 40 cycles of 95 °C for 15 s and 60 °C for 1 min. The TaqMan probe and primer sets for the target genes used in this study are shown in Table 2.

GAPDH was used as the reference gene. Ct values for GAPDH mRNA of an individual PBMC per sample were calculated. The mean was calculated from experiments performed in duplicate. For data analysis, the StepOne software v2.1 was used. Data representing the relative expression of detected mRNA normalized to GAPDH mRNA was used as a calibrator for comparative analysis. RT-qPCR was performed in accordance with the Minimum Information for Publication of Quantitative Real-Time PCR Experiments (MIQE) guidelines [19,20]. The relative expression data was calculated according to the 2^-ΔΔCt^ method.

### 2.7. Statistical Analyses

Results were analyzed using the JMP v12 statistical package (SAS Institute, Cary, NC, USA). The normality of distribution was assessed using the Shapiro–Wilk test. As mRNA levels of each TRP channel were all not normally distributed, statistical analyses were performed using Mann–Whitney U and Kruskal–Wallis H tests and correlation analysis was performed using Spearman’s rank correlation test. The Bonferroni-corrected Mann–Whitney U test was used to evaluate inter-group comparisons of the mean differences according to their distribution. Data are shown as the mean ± standard deviation (SD) or as correlation coefficients.

## 3. Results

### 3.1. TRP Channel Levels in IBD

The expression of TRP channels in PBMCs from the IBD and control groups are summarized in Figure 1. The observed fold changes in all TRP channel members were very small. However, when compared with mRNA expression levels in PBMCs from healthy controls (TRPV2, 1.54 ± 0.44; TRPV3, 0.15 ± 0.08; TRPV4, 0.95 ± 0.39; TRPM2, 1.54 ± 0.63; TRPC1, 1.73 ± 0.64), those of TRPV2 and TRPC1 were lower in both UC (1.11 ± 0.39 (0.72-fold), *p* < 0.0001 and 1.15 ± 0.65 (0.66-fold), *p* = 0.0002, respectively) and CD (1.18 ± 0.34 (0.77-fold), *p* = 0.0014 and 1.24 ± 0.65 (0.72-fold), *p* = 0.0021, respectively) groups, those of TRPM2 were higher in both UC (2.20 ± 0.76 (1.43-fold), *p* < 0.0001) and CD (2.28 ± 0.86 (1.47-fold), *p* < 0.0001) groups, those of TRPV3 were lower only in the CD group (0.09 ± 0.06 (0.58-fold), *p* = 0.001), and those of TRPV4 were higher only in the CD (1.40 ± 0.72 (1.48-fold), *p* = 0.0067) group.

Further, TRPC6 mRNA expression was higher in the CD group (1.68 ± 1.41 (1.79-fold), *p* = 0.0008) than in the UC group (0.94 ± 0.55).

In addition, the mRNA expression levels of each TRP channel in PBMCs from healthy controls were variable. We found that TRPV3 had the lowest (0.15-fold relative to GAPDH) and TRPC1 had the highest (1.73-fold relative to GAPDH) expression.

### 3.2. Relationship between TRP Channel Levels and Disease Activity

Figure 2 shows the relationship between the expression of each TRP channel member and the clinical disease progression, assessed using PMS for UC patients, and CDAI for CD patients. There was a tendency observed for the mRNA expression of TRPV2 to negatively correlate with disease activity in both UC and CD groups, while that of TRPM4 was negatively correlated with disease activity in only the UC group. However, the significance of these correlations is questionable since the R² values were <0.1.

### 3.3. Correlation between Expression of TRP Channels and Clinical Parameters

Table 3 summarizes the correlation coefficients and significance values for comparisons between the expression of each TRP channel member and the indicated laboratory parameters. In the UC group, mRNA expression of TRPV2 and TRPV3 negatively correlated with the leukocyte count, while that of TRPV4 and TRPM5 positively correlated with the serum albumin and hemoglobin levels, respectively. In the CD group, the expression of TRPV4 positively correlated with the CRP level.

### 3.4. Relationship between TRP Channel Expression and Medical Treatment

We also assessed the relationship between each TRP channel member and medical treatment received by the patient. Since the number of patients in each group was fairly small, no significant association was found between individual TRP channel members and specific medical treatments (Appendix A).

## 4. Discussion

Recently, it was revealed that TRP channels are expressed throughout the human body, including in immune cells, sensory nerves, and resident tissue cells [4,5,6,7]. Saunders et al. examined the expression of TRPV1 and TRPV2 in PBMCs from healthy subjects and speculated that as they have a role in the detection of noxious stimuli in the blood or under pathological conditions, their upregulation acts as an indicator of inflammation [13,21]. However, to the best of our knowledge, the current study is the first to explore the relationships among the gene expression profiles of TRP channel members in PBMCs from patients with IBD.

TRPV1, a polymodal receptor involved in inflammation and nociception, plays an important role in visceral hypersensitivity [22]. Reportedly, TRPV1-immunoreactivity in colonic tissue is increased, and is correlated with the severity of abdominal pain in patients with IBD [22,23]. More recently, TRPV1 was shown to be expressed in CD4+ T cells, where it regulates cell activation and proinflammatory properties in IBD [24]. Meanwhile, a TRPV1 antagonist suppressed colitis and colorectal distension in animal models [25]. In a study, two-fold upregulation of the TRPV1 gene was found in PBMCs of patients hyposensitive to capsaicin, pain, and thermal stimuli [14]. However, in the present study, we did not observe any significant differences in TRPV1 expression in PBMCs from patients with IBD compared to healthy controls. Hence, further studies are needed to determine the role of the TRPV1 pathway in PBMCs from IBD patients using protein or functional data.

TRPV2 triggers a wide range of physiological actions including changes in the innate and adaptive immune system. It is expressed in granulocytes and monocytes/macrophages and contributes to phagocytosis, migration, and inflammatory cytokine production [26]. In PBMCs, the increased expression of TRPV2 was closely correlated with childhood asthma [15]. In TRPV2-knockout mice, colitis was less severe due to the reduced infiltration of macrophages [27], suggesting that the TRPV2 pathway plays a key role in the development of colitis. Our study showed that TRPV2 expression in PBMCs decreased in patients with UC and CD and was inversely correlated with disease activity. Based on these findings, a reduction in TRPV2 expression in PBMCs may hypothetically dampen the proinflammatory response and could possibly reduce the severity of intestinal inflammation. However, further investigations are required to confirm this hypothesis.

TRPV3 is broadly expressed in intestinal epithelial cells, possibly for nutrient sensing and digestion [28,29]; however, its precise function is not well understood. With the exception of one study reporting its association with a higher risk of colorectal cancer, none others have investigated the role of TRPV3 in gut disease [30]. Interestingly, a recent study demonstrated a decrease in the proliferation rate of oral epithelial cells in TRPV3-knockout mice [31], suggesting that TRPV3 may contribute to oral wound repair. Our data revealed that TRPV3 expression was only decreased in PBMCs of patients with CD, however, its function in PBMCs requires further investigation.

TRPV4 is expressed and is functional in intestinal epithelial cells, glial cells, and CD45+ leukocytes; its activation in the gut causes increased intracellular calcium concentrations and chemokine release [28]. Studies have indicated a strong role for TRPV4 in IBD with elevated expression of TRPV4 observed in the intestinal tissue of patients with UC and CD [32]. In animal models of colitis, TRPV4 activation causes inflammation [32], and its blockade alleviates inflammation [33]. In this study, we found that the expression of TRPV4 was also upregulated in PBMCs and was correlated with the CRP level in patients with CD. Although the correlations were quite weak, and conclusions cannot be drawn based on mRNA levels without protein and modulation data, our data suggests a potential role for leukocyte TRPV4 in the pathophysiology of CD.

Recent studies have also revealed the involvement of TRPM2 in various aspects of immunity [34]. In a colitis model, TRPM2 has been implicated in inflammatory pathways, specifically as a key participant in chemokine production [35]. Contrary to this proinflammatory action, TRPM2-knockout mice exhibited decreased survival after liver infection with *Listeria monocytogenes* [36]. Thus, TRPM2 may be detrimental or beneficial depending on the underlying disease. Although no information is available on TRPM2 expression in the intestinal tissue, its upregulation in PBMCs of patients with UC and CD is of particular interest as a participant in disease pathogenesis and as a promising marker for disease activity.

TRPM4 plays a predominant role as a negative feedback mechanism during calcium oscillations, which may be important for differential gene expression in T cells [37]. A recent study showed that TRPM4 plays an important role in the immune surveillance processes. It is essential for the proper functioning of monocytes/macrophages and the efficiency of the subsequent response to infection [38], as well as the migration of dendritic cells [39]. At present, no information is available on the role of TRPM4 in gut disease. Our study showed no alteration in TRPM4 expression in PBMCs from either UC or CD patients, however, further investigation is needed.

Knowledge of the role of TRPM5, predominantly expressed by tuft cells that are an intestinal epithelial subset [40], in gut disease is limited [41]. A recent report showed that the disruption of chemosensory signaling through the loss of TRPM5 abrogates the expansion of tuft cells [42]. Interestingly, the ablation of doublecortin-like kinase 1 (DCLK1), a marker of tuft cells, in the colonic epithelium exacerbates colitis in mice [43,44]. This finding suggests that TRPM5 plays an important role in regulating the intestinal inflammatory response and epithelial integrity. Another study showed that the number of DCLK1-positive cells decreased in intestinal tissue from patients with celiac disease [45]. Meanwhile, our study found no alteration in TRPM5 expression in PBMCs from either UC or CD patients, warranting further investigation.

The TRPC subfamily comprises six members (TRPC1, TRPC3–7) in humans, many of which are ubiquitously expressed in tissues and modulate a multitude of cellular responses [46]. Of these, TRPC1, TRPC3, and TRPC6 were detectable in this study. As TRPC1 controls the release of interleukin-1 from macrophages [47] and that of tumor necrosis factor-α from mast cells [48], its decreased expression in PBMCs from patients with UC and CD seen in our study may enhance disease development. Moreover, the expression of TRPC6 was upregulated in PBMCs from patients with CD but not those with UC in this study. These results, together with those of a previous study reporting increased TRPC6 mRNA levels in stenotic areas of patients with CD [49], suggest that TRPC6 may be associated with excessive CD fibrosis.

To date, the mechanisms of TRP channel expression in PBMCs from patients with IBD remain elusive. TRP channels in PBMCs may respond to multiple stimuli present in the peripheral circulation of these patients. The different expression levels of these TRP channels may suggest a possible role as an indicator of inflammation at secondary sites, as well as involvement in IBD pathophysiology. Additionally, the difference in expression of TRPV channels in UC and CD patients may result from differences in the disease pathogenesis. Recent studies have also shown that lipopolysaccharide (LPS) activates several members of the TRP channel family, such as TRPV4 and TRPA1, as well as the Toll-like receptor 4 (TLR4), suggesting the role of TRP channels as sensors of bacterial endotoxins, and therefore, as crucial players in innate immunity. Moreover, since TRP channel and TLR expression overlap in many cell types, including immune cells and epithelial cells, it would be of interest to explore the crosstalk between intracellular signaling pathways initiated by TLR activation and TRP channel activation in patients with IBD [50]. Understanding the involvement of TRP channel members in IBD will be crucial to evaluate the potential for manipulating TRP activity as a therapeutic intervention [51,52].

As the majority of our patients were receiving medications, we assessed whether medical treatment may affect the expression of TRP channel family in PBMCs. A subgroup analysis among patients with untreated and treated IBD showed that the use of medications had no significant effect on mRNA levels of any TRP. Future studies sequentially assessing mRNA levels of TRPs in the same patient are required to confirm this lack of association.

This study had certain limitations. First, it was conducted at a single center and involved a limited number of patients, which could cause a β-error, particularly for the analysis of the medical treatment. Second, as this study analyzed the gene expression of all PBMCs, the PBMC subsets that actually express TRP channels remain to be determined. Third, this study characterized the TRP expression at the mRNA level only. To support any conclusion on the role of TRP channels in PBMCs in IBD, the evaluation of TRP channel protein levels (enzyme-linked immunosorbent assay or immunocytochemistry), as well as modulation experiments (specific activation/inhibition, knockdown/knockout in vitro, and/or in animal models) are required. Fourth, the changes in expression levels and correlation strengths observed were very small, hence, careful attention should be paid in interpreting the data. Fifth, correlations between each TRP channel and laboratory parameters did not clearly implicate functional relationships, particularly in TRPV3, TRPV4, and TRPM5 in UC, therefore, follow up studies are needed. Finally, this study analyzed leukocytes obtained from the peripheral circulation, and not from the diseased intestine. A comparison of gene expression at these two sites could help advance our understanding of the pathophysiology of IBD.

The present results indicate, for the first time, that PBMCs from patients with IBD express different mRNA levels of TRP channel members, which may play an important role in the progression of IBD. Furthermore, their expression levels in PBMCs are a promising marker for IBD. Further studies are needed to determine the clinical and pathogenic role of TRP channels in IBD.

## Figures and Tables

**Figure 1 jcm-09-02643-f001:**
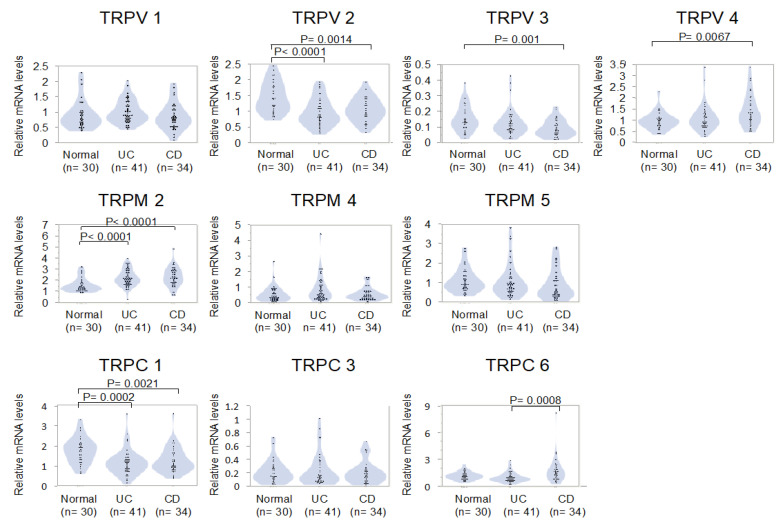
Expression of transient receptor potential (TRP) channels in healthy subjects, patients with ulcerative colitis (UC) and Crohn’s disease (CD). Bars represent mean ± SD. N: Number of subjects. Inter-group significance, Bonferroni-corrected Mann–Whitney U test, *p* < 0.01.

**Figure 2 jcm-09-02643-f002:**
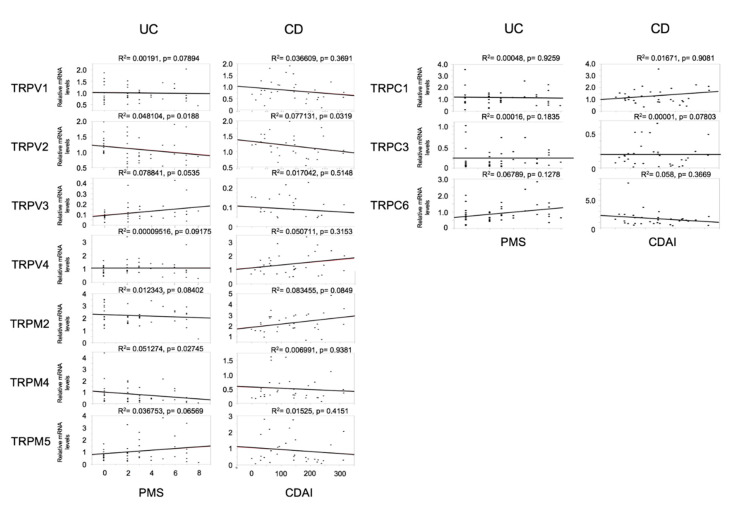
Correlation between TRP channel expression and clinical disease progression in patients with ulcerative colitis (UC) and Crohn’s disease (CD). Clinical disease progression was assessed using the partial Mayo score (PMS) for patients with UC and the CD activity index (CDAI) for patients with CD. N: Number of patients.

**Table 1 jcm-09-02643-t001:** Patient characteristics.

	Healthy Volunteers	Ulcerative Colitis	Crohn’s Disease	*p*-Value
	(*n* = 30)	(*n* = 41)	(*n* = 34)	
Sex, male/female	13/17	19/22	12/12	0.61
Age, years	39	41	35	0.17
(median, IQR)	(32–44)	(29–58)	(27–47)	
Area involved		total colitis/	ileitis/	
		left-side colitis/	ileocolitis/	
		proctitis	colitis	
		26/8/7	4/23/7	
Disease duration, months		81	106	0.19
(median, IQR)		(60.5–171.5)	(60.9–274.2)	
Treatments				
No medication		2 (4.9)	0 (0.0)	0.19
5-aminosalicylic acid (%)		36 (87.8)	25 (73.5)	0.11
Prednisolone (%)		7 (17.1)	3 (8.8)	0.3
Immunomodulator (%)		7 (17.1)	9 (26.5)	0.32
Leukocytapheresis (%)		2 (4.9)	0 (0.0)	0.19
Nutrition therapy (%)		0 (0.0)	21 (61.8)	<0.01
Anti-tumor necrosis factor (%)		3 (7.3)	24 (70.6)	<0.01
Surgery (%)		0 (0.0)	15 (44.1)	<0.01

IQR: Interquartile range.

**Table 2 jcm-09-02643-t002:** Details of TaqMan probes and primers used in this study.

Gene	Accession Number	Assay
GAPDH	NM_001256799.2	Hs02786624_g1
	NM_001289745.1	
	NM_001289746.1	
	NM_002046.5	
TRPV1	NM_018727.5	Hs00218912_m1
	NM_080704.3	
	NM_080705.3	
	NM_080706.3	
TRPV2	NM_016113.4	Hs00901648_m1
TRPV3	NM_001258205.1	Hs00376854_m1
	NM_145068.3	
TRPV4	NM_001177428.1	Hs01099348_m1
	NM_001177431.1	
	NM_001177433.1	
	NM_021625.4	
	NM_147204.2	
TRPM2	NM_001320350.1	Hs01066091_m1
	NM_001320351.1	
	NM_001320352.1	
	NM_003307.3	
TRPM4	NM_001195227.1	Hs00214167_m1
	NM_001321281.1	
	NM_001321282.1	
	NM_001321283.1	
	NM_001321285.1	
	NM_017636.3	
TRPM5	NM_014555.3	Hs00175822_m1
TRPC1	NM_001251845.1	Hs00608195_m1
	NM_003304.4	
TRPC3	NM_001130698.1	Hs00162985_m1
	NM_003305.2	
TRPC4	NM_001135955.1	Hs01077392_m1
	NM_001135956.1	
	NM_001135957.1	
	NM_001135958.1	
	NM_003306.1	
TRPC5	NM_012471.2	Hs00202960_m1
TRPC6	NM_004621.5	Hs00988479_m1
TRPC7	NM_001167576.1	Hs00220638_m1
	NM_001167577.2	
	NM_020389.2	

GAPDH: Glyceraldehyde 3-phosphate dehydrogenase; TRP: Transient receptor potential.

**Table 3 jcm-09-02643-t003:** Correlation coefficients and significance of differences between each transient receptor potential (TRP) channel expression and laboratory parameters in patients with ulcerative colitis (UC) or Crohn’s disease (CD).

		UC				CD		
	CRP	Alb	WBC	Hb	CRP	Alb	WBC	Hb
TRPV1	0.011	0.009	−0.206	0.116	−0.174	0.224	−0.275	0.0461
TRPV2	−0.111	−0.01	−0.500 **	−0.151	−0.275	0.039	−0.205	−0.199
TRPV3	0.089	0.079	−0.399 *	0.129	−0.216	0.346	−0.146	0.1235
TRPV4	0.105	0.432 *	−0.027	0.341	0.355 *	−0.229	0.167	0.1165
TRPM2	−0.282	−0.095	0.285	−0.213	0.315	−0.064	0.032	0.0448
TRPM4	−0.011	−0.03	0.161	−0.196	−0.125	0.004	−0.193	0.0211
TRPM5	−0.07	0.24	−0.013	0.378 *	−0.268	0.283	−0.344	−0.128
TRPC1	−0.125	0.031	0.033	−0.062	0.327	−0.295	0.262	−0.132
TRPC3	−0.177	−0.044	0.296	0.11	−0.182	−0.009	0.197	−0.196
TRPC6	0.038	−0.106	0.052	−0.121	−0.057	0.347	0.056	0.515 **

Alb: Albumin; CRP: C-reactive protein; Hb: Hemoglobin; WBC: White blood cell. * *p* < 0.05, ** *p* < 0.005.

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
