# Peer review of "Gene Expression of Transient Receptor Potential Channels in Peripheral Blood Mononuclear Cells of Inflammatory Bowel Disease Patients"

_jcm, 2020, doi:10.3390/jcm9082643_

Round 1
Reviewer 1 Report
In the present manuscript, Dr. Taku Morita et al have examined the expression profile of transient receptor potential (TRP) channels in peripheral blood mononuclear cells (PBMCs) from patients with inflammatory bowel disease (IBD).
TRP channels are ion channels expressed mostly on the plasma membrane on various cell types, including immune cells such as macrophages and T cells, and can modulate cytokine release, migration, or phagocytic activity. Thus, there is a rationale for studying expression of different members of the TRP superfamily in PBMC from IBD patients. In the present study, the authors have measured mRNA expression of vanilloid (TRPV1, TRPV2, TRPV3, TRPV4), melastatin (TRPM2, TRPM4, TRPM5), and canonical members (TRPC1, TRPC3, TRPC6) using Real-Time PCR in a study population of 105 subjects, including 41 patients with UC, 34 with CD, and 30 healthy subjects. The relative mRNA levels are then correlated with disease ranking (partial Mayo score (PMS) for patients with UC and the CD activity index (CDAI) for patients with CD) and laboratory parameters (total leukocyte count, serum levels of hemoglobin, albumin, and CRP). The authors conclude that PBMCs from patients with IBD exhibited varying mRNA expression levels of TRP channel members, and therefore may play an important role in the progression. The authors also write that their expression levels in PBMCs are a promising marker for IBD (line 295).
Unfortunately, the study has major limitations:
The authors only present data from mRNA-analysis and correlation data that do not implicate functional relationship. The observed fold changes in all TRP channels measured are small, and most of the correlation data are very weak or non-significant. Overall, the results presented in the manuscript are solely preliminary, and the conclusions are therefore speculative and confusing.
Specific concerns:
Figure 1: The observed fold changes in all TRP channel members are small and there seems to be large interindividual variations in the distribution. It would be more informative if individual data were visualized in e.g., violin plots. The figure legend should state how the p-values are adjusted for multiple comparisons.
Figure 2 show correlation data between expression of the 10 TRP channel genes shown in Figure 1 and clinical disease progression. The authors report that mRNA expression of TRPV2 negatively correlated with disease activity in both UC and CD groups, while mRNA expression of TRPM4 negatively correlated with disease activity in only the UC group (lines 155-156). However, with R squared values less than 0.1 (R2= 0.048, 0.077 and 0.003!) the significance of the correlations is questionable. In addition, while expression of TRPV2 are lower in UC and CD compared to healthy controls (0.72 and 0.77-fold, respectively), there is no alteration in TRPM4 expression in PBMCs from either UC or CD patients (Figure 1).
Table 3 lists correlation coefficients and significance of differences between each TRP channel mRNA expression and laboratory parameters in patients with UC or CD. Although, the five (5) significant correlations observed have more robust values (R2 between 0.378 and 0.515), correlations do not implicate functional relationship, and follow up studies are needed.
Table 4: The intention with this table is unclear to me. The authors report significant correlations between the expression of TRPV1 and that of TRPM2, TRPM4, and TRPM5, as well as between TRPM2 and TRPM4 expression levels, and conclude that this is probably due to their interference with each other’s functions (line 270). What does this mean in the context of IBD? How are these results related to the other results presented in the manuscript?
In the discussion (line 282-292), the authors themselves describe the main limitations of the study:
“First, this study was conducted at a single center and involved a limited number of patients. Second, as this study analyzed the gene expression of all PBMCs, the PBMC subsets that actually express TRP channels remain to be determined. Third, this study characterized the TRP expression at mRNA level only. To support any conclusion on the role of TRP channels in PBMCs in IBD, evaluation of TRP channel protein levels (enzyme-linked immunosorbent assay or immunocytochemistry), as well as modulation experiments (specific activation/inhibition, knockdown/knockout in vitro and/or in animal models) are required. Fourth, the change in expression levels or correlation strengths observed in this study was very small. Careful attention should be paid in interpreting the data. Finally, this study analyzed leukocytes obtained from the peripheral circulation, and not from the diseased intestine”
I agree with the authors that “Further studies are needed to determine the clinical and pathogenic role of TRP channels in IBD (line 296). Based on the results in the present manuscript, TRPV2 could be a candidate for follow up studies. Dr. Taku Morita et al suggest that a reduction in TRPV2 expression in PBMCs might hypothetically dampen the proinflammatory response and could possibly reduce the severity of intestinal inflammation (lines 2011-2015). To further investigate this hypothesis, they should perform some of the experiment suggested in the discussion.
Author Response
Reviewer#1
In the present manuscript, Dr. Taku Morita et al have examined the expression profile of transient receptor potential (TRP) channels in peripheral blood mononuclear cells (PBMCs) from patients with inflammatory bowel disease (IBD).
TRP channels are ion channels expressed mostly on the plasma membrane on various cell types, including immune cells such as macrophages and T cells, and can modulate cytokine release, migration, or phagocytic activity. Thus, there is a rationale for studying expression of different members of the TRP superfamily in PBMC from IBD patients. In the present study, the authors have measured mRNA expression of vanilloid (TRPV1, TRPV2, TRPV3, TRPV4), melastatin (TRPM2, TRPM4, TRPM5), and canonical members (TRPC1, TRPC3, TRPC6) using Real-Time PCR in a study population of 105 subjects, including 41 patients with UC, 34 with CD, and 30 healthy subjects. The relative mRNA levels are then correlated with disease ranking (partial Mayo score (PMS) for patients with UC and the CD activity index (CDAI) for patients with CD) and laboratory parameters (total leukocyte count, serum levels of hemoglobin, albumin, and CRP). The authors conclude that PBMCs from patients with IBD exhibited varying mRNA expression levels of TRP channel members, and therefore may play an important role in the progression. The authors also write that their expression levels in PBMCs are a promising marker for IBD (line 295).
Unfortunately, the study has major limitations:
The authors only present data from mRNA-analysis and correlation data that do not implicate functional relationship. The observed fold changes in all TRP channels measured are small, and most of the correlation data are very weak or non-significant. Overall, the results presented in the manuscript are solely preliminary, and the conclusions are therefore speculative and confusing.
Specific concerns:
Figure 1: The observed fold changes in all TRP channel members are small and there seems to be large interindividual variations in the distribution. It would be more informative if individual data were visualized in e.g., violin plots. The figure legend should state how the p-values are adjusted for multiple comparisons.
Response: Based on this suggestion, we have generated individual data for Figure 1 using violin plots. We have also adjusted the data in Figure 1 for multiple comparison using the Bonferroni‐corrected Mann–Whitney U test to apply a lower critical level for concluding a significant difference. We have also revised the p-value cut off to 0.01, instead of 0.05 since this case is a set of data with three tests, and have described this analysis in the figure legend. We have accordingly revised the associated text in the Results and Discussion sections.
Figure 2 show correlation data between expression of the 10 TRP channel genes shown in Figure 1 and clinical disease progression. The authors report that mRNA expression of TRPV2 negatively correlated with disease activity in both UC and CD groups, while mRNA expression of TRPM4 negatively correlated with disease activity in only the UC group (lines 155-156). However, with R squared values less than 0.1 (R2= 0.048, 0.077 and 0.003!) the significance of the correlations is questionable. In addition, while expression of TRPV2 are lower in UC and CD compared to healthy controls (0.72 and 0.77-fold, respectively), there is no alteration in TRPM4 expression in PBMCs from either UC or CD patients (Figure 1).
Response: As suggested, the significance of the correlations is questionable since R2 values are less than 0.1. Therefore, we have rewritten the description in the result section as follows on page 6, lines 151-155: “There was a tendency observed for the mRNA expression of TRPV2 to negatively correlate with disease activity in both UC and CD groups, while that of TRPM4 was negatively correlated with disease activity in only the UC group. However, the significance of these correlations is questionable since R2 values were < 0.1.”
Table 3 lists correlation coefficients and significance of differences between each TRP channel mRNA expression and laboratory parameters in patients with UC or CD. Although, the five (5) significant correlations observed have more robust values (R2 between 0.378 and 0.515), correlations do not implicate functional relationship, and follow up studies are needed.
Response: As suggested, although, the five significant correlations observed in Table 3 have more robust values, correlations between each TRP channels and laboratory parameters did not clearly implicate functional relationship particularly in TRPV3, TRPV4 and TRPM5 of UC and follow up studies are needed. We have described this in the limitations section of the revised discussion on page 9, line 285-287.
Table 4: The intention with this table is unclear to me. The authors report significant correlations between the expression of TRPV1 and that of TRPM2, TRPM4, and TRPM5, as well as between TRPM2 and TRPM4 expression levels, and conclude that this is probably due to their interference with each other’s functions (line 270). What does this mean in the context of IBD? How are these results related to the other results presented in the manuscript?
Response: As suggested by the reviewer, the conclusion made in the original manuscript stating that the results were likely due to their interference with each other’s functions, is highly speculative and unclear in the context of IBD. Therefore, we have chosen to delete this table and the associated description in the main text.
In the discussion (line 282-292), the authors themselves describe the main limitations of the study:
“First, this study was conducted at a single center and involved a limited number of patients. Second, as this study analyzed the gene expression of all PBMCs, the PBMC subsets that actually express TRP channels remain to be determined. Third, this study characterized the TRP expression at mRNA level only. To support any conclusion on the role of TRP channels in PBMCs in IBD, evaluation of TRP channel protein levels (enzyme-linked immunosorbent assay or immunocytochemistry), as well as modulation experiments (specific activation/inhibition, knockdown/knockout in vitro and/or in animal models) are required. Fourth, the change in expression levels or correlation strengths observed in this study was very small. Careful attention should be paid in interpreting the data. Finally, this study analyzed leukocytes obtained from the peripheral circulation, and not from the diseased intestine”
I agree with the authors that “Further studies are needed to determine the clinical and pathogenic role of TRP channels in IBD (line 296). Based on the results in the present manuscript, TRPV2 could be a candidate for follow up studies. Dr. Taku Morita et al suggest that a reduction in TRPV2 expression in PBMCs might hypothetically dampen the proinflammatory response and could possibly reduce the severity of intestinal inflammation (lines 2011-2015). To further investigate this hypothesis, they should perform some of the experiment suggested in the discussion.
Response: Based on the reviewer’s comment, we have revised the text as follows on page 7, line 207:
“However, further investigations are required to confirm this hypothesis.”

Reviewer 2 Report
In this manuscript the authors examined the expression profile of transient receptor potential (TRP) channels in peripheral blood mononuclear cells (PBMCs) from patients with ulcerative colitis (UC), Crohn’s disease (CD) and normal subjects.
In the section discussion the authors should specify if there are data on a potential relationship between TRP and microbiota in IBD patients. This could permit to made constructive hyptheses.
Among the limitations the authors should add the difference among the sample sizes included (41 UC, 34 CD, 30 subjects). Although the difference appears minimal it could a beta-error in the analysis of results (for example reporting on therapy).
The references should be updated, for example te reference number 2 (2006), could be replaced with the updated History of Inflammatory Bowel Diseases.
J Clin Med. 2019 Nov 14;8(11):1970. doi: 10.3390/jcm8111970.Author Response
Reviewer #2
In this manuscript the authors examined the expression profile of transient receptor potential (TRP) channels in peripheral blood mononuclear cells (PBMCs) from patients with ulcerative colitis (UC), Crohn’s disease (CD) and normal subjects.
In the section discussion the authors should specify if there are data on a potential relationship between TRP and microbiota in IBD patients. This could permit to made constructive hyptheses.
Response: Based on the reviewer’s advice, we have included the following text in the revised Discussion on page 8, line 262-268:
“Recent studies have also shown that lipopolysaccharide (LPS) activates several members of the TRP channel family, such as TRPV4 and TRPA1, as well as the Toll-like receptor 4 (TLR4), suggesting the role of TRP channels as sensors of bacterial endotoxins, and therefore, as crucial players in innate immunity. Moreover, since TRP channel and TLR expression overlap in many cell types, including immune cells and epithelial cells, it would be of interest to explore the crosstalk between intracellular signaling pathways initiated by TLR activation and TRP channel activation in patients with IBD [Ref #50]”.
Among the limitations the authors should add the difference among the sample sizes included (41 UC, 34 CD, 30 subjects). Although the difference appears minimal it could a beta-error in the analysis of results (for example reporting on therapy).
Response: Based on the reviewer’s advice, we have revised the limitations section of the manuscript to include the following: “First, it was conducted at a single center and involved a limited number of patients, which could cause a β-error, particularly for the analysis of the medical treatment.”(page 9, line 276-278)
The references should be updated, for example te reference number 2 (2006), could be replaced with the updated History of Inflammatory Bowel Diseases.
Response: Based on the reviewer’s advice, we updated the references (Ref #1 and #2)
